# Retinopathy Phenotypes in Type 2 Diabetes with Different Risks for Macular Edema and Proliferative Retinopathy

**DOI:** 10.3390/jcm9051433

**Published:** 2020-05-12

**Authors:** Ines P. Marques, Maria H. Madeira, Ana L. Messias, Torcato Santos, António C-V. Martinho, João Figueira, José Cunha-Vaz

**Affiliations:** 1AIBILI—Association for Innovation and Biomedical Research on Light and Image, 3000-548 Coimbra, Portugal; ipmarques@aibili.pt (I.P.M.); mhmadeira@aibili.pt (M.H.M.); torcato@aibili.pt (T.S.); ahmartinho@gmail.com (A.C.-V.M.); joaofigueira@oftalmologia.co.pt (J.F.); 2Dentistry Department, Faculty of Medicine University of Coimbra, 3000-075 Coimbra, Portugal; ana.messias@uc.pt; 3Department of Ophthalmology, Centro Hospitalar e Universitário de Coimbra (CHUC), 3000-075 Coimbra, Portugal

**Keywords:** diabetes, retinopathy, macular edema, proliferative retinopathy

## Abstract

Our group reported that three diabetic retinopathy (DR) phenotypes: A, characterized by low microaneurysm turnover (MAT < 6) and normal central retinal thickness (CRT); B, low MAT (<6) and increased CRT, and C, high MAT (≥6), present different risks for development of macular edema (DME) and proliferative retinopathy (PDR). To test these findings, 212 persons with type 2 diabetes (T2D) and mild nonproliferative retinopathy (NPDR), one eye per person, were followed for five years with annual visits. Of these, 172 completed the follow-up or developed an outcome: PDR or DME (considering both clinically significant macular edema (CSME) and center-involved macular edema (CIME)). Twenty-seven eyes (16%) developed either CSME (14), CIME (10), or PDR (4), with one eye developing both CSME and PDR. Phenotype A showed no association with development of vision-threatening complications. Seven eyes with phenotype B and three with phenotype C developed CIME. Phenotype C showed higher risk for CSME development, with 17.41 odds ratio (*p* = 0.010), compared with phenotypes A + B. All eyes that developed PDR were classified as phenotype C. Levels of HbA_1c_ and triglycerides were increased in phenotype C (*p* < 0.001 and *p* = 0.018, respectively). In conclusion, phenotype C identifies eyes at higher risk for development of CSME and PDR, whereas phenotype A identifies eyes at very low risk for vision-threatening complications.

## 1. Introduction

Diabetic retinopathy (DR) is one of the major complications of diabetes and leading cause of vision loss and blindness in the world, particularly among working-age adults in the United States [1]. This serious ophthalmic condition creates significant disabilities that threatens independence and impact on life quality [2]. The two major vision-threatening complications, diabetic macular edema (DME) and proliferative diabetic retinopathy (PDR) develop only in a limited number of individuals [3].

It is generally accepted that diabetes duration and level of metabolic control play a role. Still, these risk factors per se cannot explain the variability observed in DR evolution in diabetic individuals [4]. Whereas several individuals with diabetes do not develop vision-threatening retinal changes, maintaining good visual acuity after many years of diagnosis, others show progression of DR even after only a few years of diabetes, leading to vision loss.

Our group has reported on two-year and three-year follow up studies of people with type 2 diabetes (T2D) and mild nonproliferative DR (NPDR) and found marked individual variations in the progression of DR and development of complications [5,6,7,8,9]. Namely, using noninvasive methods, we have identified three different phenotypes of NPDR, based on microaneurysm turnover (MAT) and central retinal thickness (CRT), that appear to be related with different risks for the development of clinically significant macular edema. Briefly, Phenotype A is characterized by low MAT (<6) and normal CRT; Phenotype B by low MAT (<6) and increased CRT; Phenotype C by higher MAT (≥6) with or without increased CRT.

This new study is a prospective five-year study of a large cohort of people with T2D and with mild NPDR examining disease progression to vision-threatening complications, using only non-invasive examination methodologies that are easily used in clinical practice, digital color fundus photography (CFP), and optical coherence tomography (OCT), to test if different DR phenotypes show different risks for development of vision-threatening complications.

## 2. Methods

A prospective longitudinal observational cohort study (ClinicalTrials.gov identifier: NCT03010397) was designed to follow people with T2D with mild NPDR (one eye per person), graded 20 or 35 on Early Treatment Diabetic Retinopathy Study (ETDRS classification) grading scale [10]. This study is an extension of three previous studies (NCT0114599 [9], NCT01607190 [7], and EudraCT2012-001200-38 [11]). The patients in the study here reported were included according to specified inclusion and exclusion criteria and they were followed for a period of five years or until the time of development of PDR or DME, considering clinically significant macular edema (CSME) and center involved macular edema (CIME). The tenets of the Declaration of Helsinki were followed, and the Institutional Ethical Review Board approved the study. Each participant signed a written informed consent, agreeing to participate in the study, after all procedures were explained.

A total of 212 people with T2D were included, men (68.4%) and women (31.4%) with diagnosed adult-onset of T2D, aged 42 to 82 years, with an average duration diabetes of 14.1 ± 7.4 years and a mean range of hemoglobin A_1c_ (HbA_1c_) levels of 7.47 ± 1.27 (Table 1).

The study exclusion criteria comprised the presence of age-related macular degeneration, glaucoma, vitreomacular disease, high ametropia (spherical equivalent greater than -6 and +2 D), any previous laser treatment or intravitreal injections, or any patient comorbidity likely to affect the eye and not related with diabetes or cardiovascular disease. Excluded were also people with T2D with uncontrolled systemic hypertension (values outside normal range: systolic 70–210 mmHg and diastolic 50-120 mmHg), HbA_1c_ levels above 10%, during the first 6 months of the study, and a history of ischemic heart disease. Eyes with baseline central thickening identifying CIME, defined as a retinal thickness (RT) ≥ 290 μm in women and ≥ 305 μm in men [12], were also excluded.

At baseline visit (V0), demographics such as age, duration of diabetes, co-morbidities, and concomitant medication were collected for each participant. Physical assessment with biometric measures (body weight and height) and blood pressure evaluation were performed by an experienced nurse, as well as blood tests for determination of HbA_1c_ and lipid profile.

Visual acuity (best corrected visual acuity, BCVA) was measured for each eye using the ETDRS protocol and Precision Vision charts at 4 m [13].

The DR severity level was determined by two independent graders within the context of an experienced reading center and was based on the seven-field protocol following the ETDRS classification, performed by the Coimbra Ophthalmology Reading Center (CORC) [10]. One eye per person with T2D was selected at baseline as the study eye based on the inclusion/exclusion criteria. When both eyes fulfilled the criteria, the eye showing the more advanced ETDRS grading in any given individual was chosen to be the study eye.

Study follow up visits were performed at 6 months (V1), 12 months (V2), 24 months (V3), 36 months (V4), 48 months (V5), and 60 months (V6) or last visit before treatment (in the eyes that developed either CSME or PDR). At all study visits, participants underwent a complete eye examination, which included BCVA, slit-lamp examination, intraocular pressure measurement, and OCT.

During the period of the study and outside of the study visits, participants were followed in our institution in accordance with usual clinical practice.

The outcomes in the study here presented were CIME, CSME, and PDR. Center involved macular edema is defined as CRT ≥ 290 μm in women and ≥ 305 μm in men (Zeiss Cirrus SD-OCT), according to pre-defined OCT values [12]; CSME was identified on clinical examination as defined by the Early Treatment Diabetic Retinopathy Study group as retinal thickening within 500 µm of the center of the fovea or presence of hard exudates (with thickening of the adjacent retina) within 500 µm of the center of the fovea, or thickening of at least 1 disc area located less than 1 disc diameter from the center of the fovea [14]. Finally, PDR was identified by the presence of abnormal new vessels in the retina.

Laboratory analyses included HbA_1C_ concentration, glucose, creatinine, and red blood cell count, white blood cell count, platelet amount, and packed cell volume. Plasma concentrations of lipid fractionation identifying total cholesterol, high-density lipoprotein (HDL), low-density lipoprotein (LDL), and triglycerides, were measured to assess metabolic control

The calculation of the sample size was based on previous studies [6,7,8,9], where the existence of three distinct phenotypes of DR progression in T2D was proposed. Their incidence on the DR population was 50% for phenotype A, 30% for phenotype B, and 20% for phenotype C. Individuals with phenotypes B and C were shown to be at higher risk of developing CSME needing treatment, with 11% of phenotype B and 30% of phenotype C developing CSME in a two-year interval. In order to have at least five eyes from phenotype B and 10 eyes from phenotype C developing CSME needing treatment, a total of 200 eyes were considered to be needed for the study.

Baseline characteristics of the study population are presented in Table 1.

### 2.1. Color Fundus Photography

Color fundus photography (CFP) was performed according to the ETDRS protocol. The seven-fields photograph were obtained at 30/35º, using a Topcon TRC 50DX camera (Topcon Medical Systems, Tokyo, Japan), with a resolution of 3596 × 2448 pixels for ETDRS DR classification at CORC, according to the ETDRS grading scale [10].

Additionally, 45/50º 2-field images were obtained and subjected to automated microaneurysm (MA) analyses using the RetmarkerDR (Retmarker SA, Coimbra, Portugal). This automated computer-aided diagnostic system is a software that allows earmarking MA and red dot-like vascular lesions in the macula (all referred to as MA); it includes an algorithm co-registration that facilitates the comparison within the same retinal location between different visits for the same eye [15,16], as previously described [6]. Briefly, the algorithm computes for each eye the number of MAs in each visit, and the number of MAs that appear and/or disappear from one visit to the other, allowing a calculation of the number of MAs appearing and/or disappearing per time interval (i.e., the MA formation rate and the MA disappearance rate, respectively). The microaneurysm turnover (MAT) is computed as the sum of the MA formation and disappearance rates.

### 2.2. Optical Coherence Tomography

OCT was performed using the Cirrus Zeiss 5000 AngioPlex (Carl Zeiss Meditec, Dublin, CA, USA).

The Macular Cube 512 × 128 acquisition protocol, consisting of 128 B-scans with 512 A-scans each, was used to assess the subjects’ average central retinal thickness (CRT). Retinal layers segmentation for layer thickness calculation was performed on the structural OCT or OCTA (in the last visit) using the segmentation software implemented by AIBILI [17]. In the first step of our implementation, the algorithm segments large voxel intensity variations, i.e., vitreous to Retinal Nerve Fiber Layer (RNFL), Inner Segment (IS) to Outer Segment (OS), and Retinal Pigment Epithelium (RPE) to choroid interfaces. The two outer most interfaces are then used as boundaries to find the remaining inner retinal layer interfaces. All surface interfaces are then smoothed by cubic splines. This method is able to segment the OCT volume into seven retinal layers, namely, RNFL, Ganglion Cell Layer and Inner Plexiform Layer (GCL + IPL), Inner Nuclear Layer (INL), Outer Plexiform Layer (OPL), Outer Nuclear Layer and Inner Segment (ONL + IS), OS, and RPE. Automated analysis results were reviewed by a masked grader. 

Eyes with CIME were identified in agreement with the reference values established by the DRCR.net for Cirrus SD-OCT (retinal thickness greater than or equal to 290 µm in women and 305 µm in men [12]). Retinal Nerve Fiber Layer (RNFL) and/or GCL + IPL thickness decreases were considered to identify neurodegeneration [18], whereas full retina thickness increases were considered to identify edema [9], comparing to a healthy control population [17].

Data was also collected for OCT analysis of retinal segmentations in the inner and outer-rings, OCT-leakage [19], and OCT-Angiography [20]. OCT-A was performed only in the last two annual visits and the data collected could not be analyzed in correlation with the outcomes CSME, CIME or PDR.

### 2.3. Characterization of DR Phenotypes

The three different DR phenotypes for NPDR, previously described by our group [3,6], were identified according to the following rules: Phenotype A: MAT < 6 and normal RT values (CRT < 220 µm, i.e., normal mean ±1 SD); Phenotype B: MAT < 6 and increased RT values (CRT ≥ 220 µm); Phenotype C: MAT ≥ 6, with or without increased CRT.

### 2.4. Statistical Analysis

Data on each eye/patient is represented as means and corresponding standard deviations for continuous variables or absolute and relative frequencies for categorical and ordinal variables. Accordingly, a comparison of baseline characteristics of the cohorts was performed using Mann-Whitney test (due to violation of assumption of normality) or the Chi-square test with Monte-Carlo correction.

Univariate logistic regression models were used to determine the odds ratio of developing the outcomes, CSME and CIME, associated with each demographic (age, diabetes duration, gender, BMI), systemic (HbA_1C_, total cholesterol, HDL, LDL, triglycerides, systolic blood pressure and diastolic blood pressure) and ocular (MAT, MA formation rate, MA disappearance rate, CRT and GCL + IPL parameters) variables. A multivariate analysis, considering the set of systemic and non-collinear ocular parameters with *p*-values ≤ 0.10, was performed to determine the adjusted odds ratio for the development of outcomes for people with T2D categorized with phenotypes B and C. A multivariate logistic regression with block entry of the systemic and non-collinear ocular parameters/variables that presented *p*-values ≤ 0.10 in the univariate analysis was performed to model the adjusted odds ratio for the development of outcomes for people with T2D categorized with phenotypes B and C.

All statistical analyses were performed with IBM^®^ SPSS^®^ Statistics version 24.0 (IBM Corp ©, New York, NY, USA), and *p*-values < 0.05 were considered statistically significant. 

## 3. Results

From the 212 eyes included in the study, 172 completed the five-year follow-up or achieved one of the outcomes: CSME, CIME, or PDR (Figure 1). Forty participants dropped out of the study (nine died, 10 were lost to follow-up, and 21 chose to withdraw from the study).

The eyes included in the study had only mild DR, with 58 (27%) graded as ETDRS level 20 and 154 (73%) graded as ETDRS level 35. No statistically significant differences were found at baseline between the 172 eyes/patients that reached the study outcome or that performed the last visit of the study (five-year visit) and the 40 eyes from persons with T2D that dropped out of the study (Table 1).

Characterization of the eyes according to the different phenotypes previously described, was performed at the six-month visit. There were 66 (38%) eyes categorized as phenotype A, 50 (29%) eyes identified as phenotype B and 56 (33%) eyes as phenotype C within the people with T2D that completed the study, indicating a similar distribution to baseline (*p* = 0.980) and no attrition bias due to loss of follow-up (Table 1).

The demographic, systemic and ocular characteristics of each phenotype at baseline (except MAT, which was defined at six-month visit) are described in Table 2. No significant differences in age, duration of diabetes, sex, blood pressure levels, body mass index, and blood lipid levels, with the exception of triglycerides, could be found between the three different phenotypes at baseline. There was a statistically significant difference in HbA_1c_ between groups (*p* < 0.0001), attributable to higher levels of HbA_1c_ in Phenotype C when compared to phenotypes A (*p* = 0.023) and B (*p* < 0.0001). Baseline values of triglycerides were also not similar across phenotypes (*p* = 0.018), which was attributable to higher values associated with Phenotype C when compared to phenotype B (*p* = 0.014).

As a consequence of the definition criteria of phenotypes, MAT values were significantly increased in phenotype C (*p* < 0.0001). Accordingly, central subfield retinal thickness was increased in phenotype B when compared to both phenotypes A and C (*p* < 0.0001). Phenotype C also presented significantly higher retinal thickness than phenotype A. No significant alterations were observed in the GCL + IPL thickness in the inner ring between different phenotypes.

Phenotype C was identified mainly in eyes graded with ETDRS level 35 (97%), when compared with the other phenotypes, showing its association with the more advanced stage of the retinal disease included in the study.

Over the five-year follow-up period, of the eyes with phenotype A there was two-step ETDRS grade improvement in 9% (6/66 eyes) and two-step worsening in only 3% (2/66). Phenotype B showed two-step ETDRS grade improvement in 10% (5/50) with no eyes showing two-step ETDRS grade worsening. In eyes with phenotype C, only one eye showed two-step ETDRS grade improvement, whereas 23% (13/56 eyes) showed two-step ETDRS grade worsening. Twenty-seven eyes developed vision-threatening complications during the period of five-year follow-up: CSME in 14, CIME in 10, and PDR in four, with one eye/patient developing both CSME and PDR. The distribution of these outcomes in the different phenotypes is depicted on Table 3.

Clinically significant macular edema developed in three eyes (21%) categorized as phenotype B, and in 11 eyes (79%) categorized as phenotype C, corresponding to a crude odds ratio of 9.97 (2.64–37.62) for CSME development in phenotype C. The univariate logistic regression analysis also determined younger age, higher HbA_1c_, and higher LDL cholesterol as systemic significant risk factors for the development of CSME. All ocular parameters are identified as significant in the determination of the risk of development of CSME (Table 4). The multivariate logistic regression analysis, considering all those variables potentially influencing the risk associated of development of CSME (*p* ≤ 0.10 in univariate analysis) determined an adjusted OR for phenotype C of 17.41 (1.98–153.00), *p* = 0.010. Only age, BMI and GCL + IPL CSF changes further contributed to increase the overall prediction of cases. The model presented an accuracy in classification of 95.4%, with 99.3% specificity. Despite the sensitivity of 57.1%, the model positively predicted 88.9% of the cases.

Center involved macular edema developed in seven eyes (70%) categorized as phenotype B and in three eyes identified as phenotype C, indicating an OR of 6.13 (1.51–24.85) for the first. The multivariate adjustment of the OR for the development of CIME was unable to detect a statistically significant impact of phenotype B (*p* = 0.223).

Eyes with phenotype A did not develop macular edema during the five-year period of follow-up. Finally, PDR developed only in eyes categorized as phenotype C.

Analysis of the time to event of the development of outcomes, CIME, CSME, and PDR, along the five–year period, is shown in Figure 2A. There was, generally, a progressive decline in the number of cases presenting CIME, suggesting that their occurrence is not correlated with the duration of the disease but rather an individual response associated with retinal thickness baseline values. The predominance of complications in phenotype C is well demonstrated in Figure 2B.

## 4. Discussion

This five–year longitudinal study of a relatively large cohort of people with T2D and mild DR (ETDRS gradings 20 and 35) confirms and extends previous studies by our group [6,7,8,9] showing that there are different DR phenotypes with different risks for the development of vision–threatening complications, CSME, CIME, and PDR.

Using only non–invasive procedures, easy to use repeatedly in clinical practice, the study shows that characterization of different NPDR phenotypes indicate that the chance of developing CSME and PDR within a period of five years, is much higher if people with T2D have an increased MAT (≥ 6 at six months using the RetmarkerDR), which identifies phenotype C. In fact, our data suggests that Phenotype C is associated with an increased likelihood of development of CSME, whereas phenotype B is mainly associated with likelihood of development of CIME. These findings suggest that different factors underly the development of CIME and CSME. The association of increased values of MAT with CSME points to the potential relevance of capillary closure and ischemia in the process of CSME development. Therefore, CIME may not be necessarily a predictive factor for CSME [21]. It is also an argument against relying entirely on OCT metrics to identify DME [4]. Of major relevance for clinical management of diabetic retinal disease is the observation that eyes with phenotype A, which is characterized by low MAT and no increase in retinal thickness, representing approximately 40% of the mild NPDR population enrolled in the study, did not develop any vision–threatening complication, CSME, CIME, or PDR, during the five–year period of follow–up. This observation is highly noteworthy, confirming our previous observations [6]. It indicates that a large proportion of eyes presenting already DR will progress very slowly and are not likely to develop vision–threatening complications in a period of five years. Phenotype A is also the phenotype associated with increased thinning of the GCL + IPL in the central subfield. This observation contrasts with the absence of changes in the GCL + IPL in the inner ring, outside the central subfield.

It is also important to register that the relative proportion of the different phenotypes remains similar in the different studies [6,7,8,9]. People with T2D with initial stages of NPDR present phenotype A in 40% to 50%, whereas the remaining 50% are distributed between phenotypes B and C depending on other factors such as ethnicity [22].

Phenotype C was identified mainly in eyes with ETDRS grade 35 suggesting that ETDRS grade 35 may be the turning point in the progression of DR. Eyes with ETDRS grade 35 apparently reach a status of microvascular damage that creates the conditions for either stabilization or progression demonstrated by identification of Phenotype C. In this study, approximately 44% of the eyes graded as ETDRS 35 could be classified as phenotype C. Noteworthy, the eyes with phenotypes A and B remained mostly stable through the five–year follow–up, with only two eyes with phenotype A (3%) and none with phenotype B presenting two–step ETDRS grade worsening, whereas phenotype C showed two–step ETDRS worsening in 13 eyes (23%).

A limitation of this study is the focus on the initial stages of DR, allowing conclusions to be made only on the development of vision threatening complications of people with T2D with ETDRS levels 20 and 35. Furthermore, the population studied is relatively well–controlled, chosen using exclusion criteria such as excessive HbA_1C_ levels (>10%) and uncontrolled blood pressure. However, the use of these criteria guaranteed a relatively homogenous population. Another possible limitation is the relatively small number of people with T2D that completed the five–year period of follow–up. However, the five years duration of the study is of major value and offers new insights into the progression of retinal diabetic disease.

The findings here reported have clear implications on clinical trial design to test new therapies to stop progression of DR. The development of an effective drug must take into account the need to demonstrate efficacy in the early and reversible stages of diabetic retinal disease by demonstrating its effect on surrogate endpoints which can be followed for relatively short period of time. The five–year period of follow–up of our study offers information that is crucial for such designs. This study also shows that noninvasive methodologies can be used to identify the eyes that are at risk of progression and develop vision–threatening complications. According to these observations, only eyes with phenotype C should be considered for inclusion in a clinical study expected to run for less than five years, particularly if the agent to be tested is directed at the prevention of vision–threatening complications of diabetic retinal disease.

Finally, the observations here reported offer promising perspectives for personalized management of DR. After diagnosis of NPDR and still in the initial stages of retinal disease, three different phenotypes can be identified through fundus photography, including MAT evaluation using the RetmarkerDR and OCT. These examinations are easy to perform and can be repeated easily without major inconvenience to the patient. This study confirms their value for improved management strategies of NPDR and timely diagnosis of vision–threatening complications of diabetes.

The retinopathy phenotypes identified in people with T2D show different risks for vision–threatening complications. Phenotype A is a slow progression phenotype that may not even need to be followed at short intervals, appearing that longer than one–year intervals are acceptable. On the other hand, people with T2D presenting phenotype C should receive the most attention.

## 5. Conclusions

Different retinopathy phenotypes in T2D show different five–year risks for development of CSME, CIME and PDR. Phenotype C identifies eyes at higher risk for development of vision–threatening complications (CSME or PDR). It is also the only phenotype associated with PDR. Phenotype B show higher risk for development of CIME. In contrast, phenotype A identifies eyes that are at a very low risk of development of vision–threatening complications. 

## Figures and Tables

**Figure 1 jcm-09-01433-f001:**
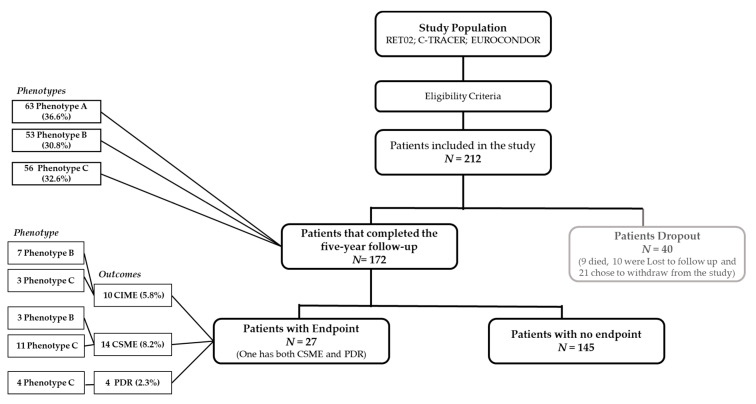
Composition of the patients included in the study over the study period: CONSORT flowchart. CIME: center-involved macular edema; CSME: clinically significant macular edema; PDR: proliferative retinopathy.

**Figure 2 jcm-09-01433-f002:**
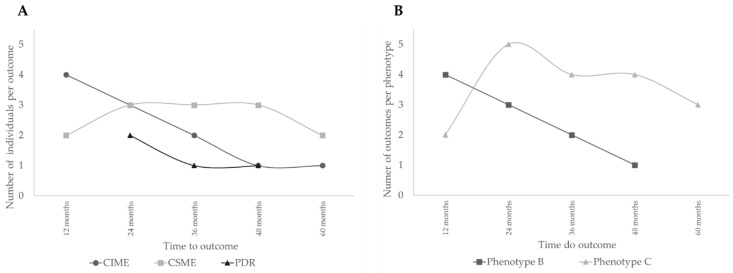
Analysis of the time to event of the development of outcomes. (**A**) Number of people with type 2 diabetes (T2D) developing different outcomes; (**B**) Number of people with T2D developing outcomes per phenotype.

**Table 1 jcm-09-01433-t001:** Demographic and clinical characteristics.

Characteristics	Included Patients *N* = 212	Study Population *N* = 172	*p*-Value
**Demographics**			
Males/Females, *N* (%)	145 (68.4)/67 (31.4)	117 (68.0)/55 (39.0)	0.865
Age, mean ± SD, year	63.0 ± 7.3	62.7 ± 7.2	0.187
Diabetes duration, mean ± SD, year	14.1 ± 7.4	14.2 ± 7.4	0.481
**Systemic characteristics**			
BMI, mean ± SD, kg/m^2^	30.1 ± 5.8	30.10 ± 5.87	0.554
Systolic BP, mean ± SD, mm Hg	137.9 ± 15.8	138.1 ± 15.9	0.874
Diastolic BP, mean ± SD, mm Hg	72.2 ± 8.6	71.9 ± 9.0	0.431
HbA1c, mean ± SD, %	7.47 ± 1.27	7.5 ± 1.3	0.182
Total cholesterol, mean ± SD, mg/dL	182.79 ± 38.40	184.02 ± 38.56	0.191
HDL cholesterol, mean ± SD, mg/dL	47.29 ± 11.65	47.45 ± 11.08	0.517
LDL cholesterol, mean ± SD, mg/dL	121.16 ± 31.69	122.24 ± 32.78	0.210
Triglycerides, mean ± SD, mg/dL	169.29 ± 116.23	166.38 ± 93.48	0.647
**ETDRS level**			
20, *N* (%)	58 (27.4)	48 (27.9)	0.359
35, *N* (%)	154 (72.6)	124 (72.1)	
**Phenotypes, *N* (%)**			
Phenotype A	84 (39.6)	66 (38.4)	0.980
Phenotype B	60 (28.3)	50 (29.1)	
Phenotype C	68 (32.1)	56 (32.6)	

**Table 2 jcm-09-01433-t002:** Characteristics of each phenotype at baseline (*n*=172).

	Phenotype A *N* = 66	Phenotype B *N* = 50	Phenotype C *N* = 56	*p*-Value
**Demographics**				
Age, year	62.83 ± 7.4	64.6 ± 6.3	61.0 ± 7.4	0.060
Duration of diabetes, year	13.4 ± 7.4	15.2 ± 8.7	14.2 ± 6.0	0.393
Males/Females, frequency (%)	44/22 (66.7/33.3)	36/14 (72.0/28.0)	37/19 (66.1/33.9)	0.808
Left eye/Right eye, frequency (%)	29/37 (43.9/56.1)	28/22 (56.0/44.0)	23/33 (41.1/58.9)	0.270
**Systemic characteristics**				
HbA1C, %	7.4 ± 1.1^a^	7.1 ± 1.2^b^	8.1 ± 1.3^a, b^	**<0.001***
Total cholesterol, mg/dL	186.2 ± 33.3	183.6 ± 36.7	181.8 ± 46.7	0.750
HDL, mg/dL	46.5 ± 9.9	49.9 ± 12.1	46.3 ± 11.3	0.172
LDL, mg/dL	124.9 ± 32.0	121.0 ± 30.4	120.2 ± 36.0	0.533
Triglycerides, mg/dL	162.6 ± 88.0	146.9 ± 88.3^a^	188.5 ± 101.3^a^	**0.018***
Systolic blood pressure, mm Hg	136.3 ± 14.2	136.8 ± 17.2	141.3 ± 16.3	0.345
Diastolic blood pressure, mm Hg	70.9 ± 7.9	71.5 ± 9.9	73.6 ± 9.1	0.248
BMI, kg/m^2^	30.9 ± 6.1	28.6 ± 5.4	30.5 ± 5.9	0.079
**Ocular characteristics**				
BCVA, letters	85.4 ± 3.7	85.7 ± 4.2	85.7 ± 4.0	0.846
MA turnover, no. per 6 months	1.9 ± 1.8^a^	2.1 ± 1.8^b^	17.5 ± 17.7^a, b^	**<0.001***
MA formation rate, no. per 6 months	0.7 ± 1.1^a^	0.7 ± 1.0^b^	8.0 ± 7.4^a, b^	**<0.001***
MA disappearance rate, no. per 6 months	1.1 ± 1.6^a^	1.4 ± 1.4^b^	9.5 ± 9.3^a, b^	**<0.001***
Central subfield RT, µm	252.0 ± 18.2^a, b^	285.6 ± 9.3^b, c^	267.3 ± 20.2^a, c^	**<0.001***
GCL+IPL CSF thickness, µm	35.1 ± 8.1^a^	44.3 ± 8.7^a, b^	39.4 ± 9.5^b^	**<0.001***
GCL + IPL InRing thickness, µm	89.8 ± 8.2	91.3 ± 12.7	91.5 ± 9.1	**0.192**
ETDRS 10-20/35, frequency (%)	23/43 (34.8/65.2)^a^	23/27 (46.0/54.0)^b^	2/56 (3.6/96.4)^a, b^	**<0.001***

* and Bold values represent statistically significant alterations, with *p* < 0.05. Similar superscript letters denote groups that differ at 0.05 level: ^a^ = Phenotype A vs. B, ^b^ = Phenotype A vs. B and ^c^ = Phenotype A vs. B. HDL: high-density lipoprotein; LDL: low-density lipoprotein; BCVA: best corrected visual acuity; MA: microaneurysm RT: retinal thickness; GCL: Ganglion Cell Layer; IPL: Inner Plexiform Layer; CSF: Central Subfield; ETDRS: Early Treatment Diabetic Retinopathy Study.

**Table 3 jcm-09-01433-t003:** Distribution of outcomes across phenotypes. *N* (% within row).

Phenotype	Outcome					
	No Outcome	CIME	CSME	PDR	Total	*p*-Value
A	66 (100%)	0 (0%)	0 (0%)	0 (0%)	66	**<0.001***
B	40 (80%)	7 (14%)	3 (6%)	0 (0%)	50	
C	39 (69.6%)	3 (5.4%)	11^#^ (19.6%)	4^#^ (7.1%)	56⁺	
Total	145 (84.3%)	10 (5.8%)	14 (8.1%)	4 (2.3%)	172	

* and Bold values represent statistically significant alterations, with *p* < 0.05. ^#^ One patient developed both CSME and PDR and is considered in the two outcomes.

**Table 4 jcm-09-01433-t004:** Univariate and multivariate analyses of OR for CSME and CIME on a logistic regression.

	Univariate Model				Multivariate Model			
	CSME		CIME		CSME ^†^		CIME ^†^	
	OR (95% CI)	*p*	OR (95% CI)	*p*	OR (95% CI)	*p*	OR (95% CI)	*p*
**Demographic characteristics**								
Age	0.87 (0.79–0.95)	**0.001***	1.01 (0.92–1.11)	0.835	0.85 (0.74–0.99)	**0.033** *****		
Diabetes duration	0.96 (0.88–1.04)	0.305	0.96 (0.87–1.05)	0.385				
Gender (female)	1.72 (0.56–5.26)	0.340	0.57 (0.12–2.81)	0.493				
BMI	0.90 (0.81–1.00)	0.051	0.90 (0.79–1.03)	0.124	0.78 (0.64–0.96)	**0.017***		
**Systemic characteristics**								
HbA_1c_	1.58 (1.02–2.43)	**0.040***	0.70 (0.39–1.24)	0.221	0.73 (0.33–1.63)	0.447		
Total cholesterol	1.01 (1.00–1.03)	0.087	0.99 (0.97–1.00)	0.106				
HDL	0.94 (0.88–1.00)	0.059	1.01 (0.06–1.07)	0.647	0.90 (0.79–1.02)	0.097		
LDL	1.02 (1.00–1.03)	**0.043***	0.98 (0.96–1.00)	0.102	1.03 (1.00–1.06)	0.061	0.99 (0.97–1.02)	0.601
Triglycerides	1.00 (1.00–1.01)	0.088	0.99 (0.99–1.01)	0.395	1.00 (0.99–1.01)	0.645		
Systolic blood pressure	0.99 (0.95–1.03)	0.552	0.96 (0.92–1.00)	0.054			0.9 4 (0.90–0.99)	**0.018***
Diastolic blood pressure	0.97 (0.91–1.04)	0.454	0.9 5 (0.88–1.03)	0.237				
**Phenotype**								
Phenotype B	0.72 (0.19–2.70)	0.622	6.13 (1.51–24.85)	**0.011***			2.82 (0.53–14.99)	0.223
Phenotype C	9.97 (2.64–37.62)	**0.001***	1.17 (0.29–4.73)	0.831	17.41 (1.98–153.00)	**0.010***		
**Ocular characteristics**								
MA turnover	1.08 (1.03–1.13)	**0.001***	0.96 (0.85–1.08)	0.507				
MA formation rate	1.16 (1.06–1.28)	**0.002***	0.70 (0.42–1.17)	0.173				
MA disappearance rate	1.13 (1.05–1.22)	**0.001***	1.00 (0.87–1.15)	0.961				
Retinal thickness at baseline	1.06 (1.03–1.11)	**0.001***	1.19 (1.09–1.31)	**<0.001***				
GCL + IPL CSF thickness	1.14 (1.05–1.22)	**0.001***	1.21 (1.09–1.35)	**<0.001***	1.20 (1.05–1.36)	**0.005***	1.24 (1.08–1.42)	**0.002***
GCL + IPL InRing thickness	1.12 (1.04–1.20)	**0.004***	1.05 (0.96–1.13)	0.283	0.98 (0.86–1.12)	0.762		
Ratio RT/GCL (CSF)	0.54 (0.32–0.90)	**0.019***	0.55 (0.30–1.00)	0.050				

* and Bold values represent statistically significant alterations, with *p* < 0.05. ^†^ Multivariate models included variables with *p* ≤ 0.1 in univariate analysis. Variables MA turnover, MA formation rate, MA disappearance rate and Retinal thickness not included due to the inherent inclusion in the definition of phenotypes. Total cholesterol and Ratio RT/CGL (CSF) not included due to multicollinearity issues with the variables that compose it.

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
