# Peer review of "Retinopathy Phenotypes in Type 2 Diabetes with Different Risks for Macular Edema and Proliferative Retinopathy"

_jcm, 2020, doi:10.3390/jcm9051433_

Round 1

Reviewer 1 Report

I read with great interest the manuscript entitled Different Retinopathy phenotypes in diabetes Type 2 show different five-year progression risks for developing macular edema and proliferative retinopathy. This manuscript presents an elegant study which in the main is very well written and adds greatly to the story and epidemiology of DR. However I think the title and abstract need some work.

Throughout the manuscript authors should consider the relevance of language matters when discussing diabetes. Terms like diabetic and patient with diabetes should be changed to people with diabetes and diabetes type 2 should be replaced with people with Type 2 diabetes.

Title needs to be changed as it doesn’t read well

Abstract

The first sentence is clumsy and doesn’t read well suggest re-writing it.

Page 1 line 21 – you state 212 patients but on line 22 you have 212 eyes/patients then line 23 you state 27, 14, 10 and 4 eyes is your analysis at the patient or individual eye level?

Lines 25-29 discusses phenotypes A-C this is the first time they are mentioned and you haven’t introduced what these refer to or how many people were in each of the phenotypes etc.

Overall I find the abstract a little confusing which considering how well the rest of the manuscript is written is odd. I would suggest a little more time is spent re-writing this.

Introduction

Diabetic retinopathy is abbreviated in the opening sentence to DR however, you switch between retinopathy and DR throughout the manuscript once abbreviated you should be consistent in the use of DR.

Methods

Page 4 line 67 I find the use eye/patients confusing are you counting individual eyes or the number of people? Please just stick to one or state xx eyes in xx people.

HbA1c should be written as HbA1c

Page 4 line 76-77 – would be useful to have %M and %F and the duration of diabetes. The mean of range of HbA1c would be better than just the maximum. Not sure of this journals rules on HbA1c reporting but most support dual reporting of mmol/mol and (%).

Line 81-82 you state any other systemic disease that could affect the eye. Could you give some examples.

At the baseline visit were co-morbidities recorded?

Page 6 line 124 – abbreviation of CFP needs brackets (CFP)

Line 125 – the seven-field photographs, no need for the s at the end of field

Line 128 – should be 2 field not field-2

Page 7 line 173 – 177 – I think the sentence is missing a word at the end possibly variables?

Line 177 – in the multivariate analysis which method was used for inclusion/exclusion of variables i.e. forward/backwards stepwise etc

Results

Figure 1 – the inclusion of this diagram is a great addition and makes it clear what happened to the patients etc.

Page 13 line 256 phenotype the c is on line 257 which is after figure 2

Of the 53 people with phenotype B only 10 developed one of the end points and of the 56 people with phenotype C only 18 developed one of the end points. Whilst it is easy to see phenotype C is the more advanced/aggressive of the 3 the authors don’t spend any time discussing the remaining 43/53 people in phenotype B or the 36/56 people in phenotype C who do not develop an end point of interest. Is there any progression of retinopathy in these cases? Could this mean there are more than 3 phenotypes with phenotype B and C needing further subclassification. Similarly was there any progression at all of retinopathy in phenotype A? I think a brief mention of these in the results with some discussion of this later on would greatly add to the study.

Author Response

Answers to Reviewer #1

I read with great interest the manuscript entitled Different Retinopathy phenotypes in diabetes Type 2 show different five-year progression risks for developing macular edema and proliferative retinopathy. This manuscript presents an elegant study which in the main is very well written and adds greatly to the story and epidemiology of DR. However I think the title and abstract need some work.

Throughout the manuscript authors should consider the relevance of language matters when discussing diabetes. Terms like diabetic and patient with diabetes should be changed to people with diabetes and diabetes type 2 should be replaced with people with Type 2 diabetes.

Title needs to be changed as it doesn’t read well

We thank to the reviewer on its general comments. The language used along the manuscript was revised, and we have also changed the title.

  • Abstract

The first sentence is clumsy and doesn’t read well suggest re-writing it.

Page 1 line 21 – you state 212 patients but on line 22 you have 212 eyes/patients then line 23 you state 27, 14, 10 and 4 eyes is your analysis at the patient or individual eye level?

Lines 25-29 discusses phenotypes A-C this is the first time they are mentioned and you haven’t introduced what these refer to or how many people were in each of the phenotypes etc.

Overall I find the abstract a little confusing which considering how well the rest of the manuscript is written is odd. I would suggest a little more time is spent re-writing this.

 The abstract has been rewritten. We followed the recommendation of the journal to avoid using the usual division: purpose, methods, results and conclusion.

  • Introduction
  • Diabetic retinopathy is abbreviated in the opening sentence to DR however, you switch between retinopathy and DR throughout the manuscript once abbreviated you should be consistent in the use of DR.

Attention has been given to the language uniformization, maintaining DR abbreviation and avoiding the use of patients, which was replaced, as suggested by “people with T2D”.

  • Methods
  • Page 4 line 67 I find the use eye/patients confusing are you counting individual eyes or the number of people? Please just stick to one or state xx eyes in xx people.

We have now clearly stated that we have one eye per patient and have used either eye or person with T2D (to avoid the use of the term patient), when referring to an individual case.

  • HbA1c should be written as HbA1c

We have corrected it along the manuscript and in the tables

  • Page 4 line 76-77 – would be useful to have %M and %F and the duration of diabetes. The mean of range of HbA1c would be better than just the maximum. Not sure of this journals rules on HbA1c reporting but most support dual reporting of mmol/mol and (%).

We have now included in this paragraph the % of females, average of diabetes duration and mean range of HbA1C, as presented in table 1. A total of 212 people with T2D were included, men (68.4%) and women (31.4%) with diagnosed adult-onset of T2D, age 42 to 82 years, with an average duration diabetes of 14.1 ± 7.4 years and a mean range of hemoglobin A1c (HbA1c) levels of 7.47 ± 1.27 (table 1).”

  • Line 81-82 you state any other systemic disease that could affect the eye. Could you give some

The sentence has been changed. It is now stated that “any patient comorbidity, likely to affect the eye and not related with diabetes or cardiovascular disease.”

  • At the baseline visit were co-morbidities recorded?

Yes, co-morbidities were recorded. We thank to the reviewer for noticing this missing case. We have included it in the manuscript.

  • Page 6 line 124 – abbreviation of CFP needs brackets (CFP)
  • Line 125 – the seven-field photographs, no need for the s at the end of field
  • Line 128 – should be 2 field not field-2

We thank the reviewer for the 3 above corrections, that were now taken in consideration in the manuscript.

  • Page 7 line 173 – 177 – I think the sentence is missing a word at the end possibly variables?

Yes, thank you. The word “variables” has been added to the end of the sentence.

  • Line 177 – in the multivariate analysis which method was used for inclusion/exclusion of variables i.e.forward/backwards stepwise etc

In the multivariate analysis the inclusion/exclusion criteria were based on a model of block entry of

of all variables that had, in the univariate analysis, p≤0.1. This method was used to decrease the model complexity and enhance its external variability. To clarify this, we included in the manuscript, the sentence (lines 193-200): “A multivariate logistic regression with block entry of the systemic and non-collinear ocular parameters/variables that presented p-values ≤ 0.10 in the univariate analysis was performed to model the adjusted odds ratio for the development of outcomes for people with T2D categorized with phenotypes B and C”.

  • Results
  • Figure 1 – the inclusion of this diagram is a great addition and makes it clear what happened to the patients etc.

Thank you.

  • Page 13 line 256 phenotype the c is on line 257 which is after figure 2

Thank you. This presentation error was corrected.

  • Of the 53 people with phenotype B only 10 developed one of the end points and of the 56 people with phenotype C only 18 developed one of the end points. Whilst it is easy to see phenotype C is the more advanced/aggressive of the 3 the authors don’t spend any time discussing the remaining 43/53 people in phenotype B or the 36/56 people in phenotype C who do not develop an end point of interest. Is there any progression of retinopathy in these cases? Could this mean there are more than 3 phenotypes with phenotype B and C needing further subclassification. Similarly was there any progression at all of retinopathy in phenotype A? I think a brief mention of these in the results with some discussion of this later on would greatly add to the study.

We thank the reviewer for this suggestion. The information on ETDRS grade progression for each phenotype on the 5-years of the study was included on lines 244-249, with the sentence “Over the 5-year follow-up period, of the eyes with phenotype A there was 2-step ETDRS grade improvement in 9% (6/66 eyes) and 2-step worsening in only 3% (2/66). Phenotype B showed 2-step ETDRS grade improvement in 10% (5/50) with no eyes showing 2-step ETDRS grade worsening. In eyes with phenotype C, only one eye showed 2-step ETDRS grade improvement, whereas 23% (13/56 eyes) showed 2-step ETDRS grade worsening.” We have also included a reference to it in the discussion (lines 322-325) “Noteworthy, the eyes with phenotypes A and B remained mostly stable through the 5-year follow-up, with only 2 eyes with phenotype A (3%) and none with phenotype B presenting 2-step ETDRS grade worsening, whereas phenotype C showed 2-step ETDRS worsening in 13 eyes (23%).”

Reviewer 2 Report

General

In their manuscript entitled “Different Retinopathy phenotypes in Diabetes Type 2 show different five-year progression risks for developing macular edema and proliferative retinopathy” Marques et al. report on results of a prospective follow-up study in which the authors observed that the initial development of diabetes, classified into three groups, correlates with further long-term disease progression.

Results are part of the study with 9 mentioned endpoints (ClinicalTrials.gov). Here only few results of the whole study are described. The authors should give information on the other endpoint data.

Grammar, style and linguistic performance of the manuscript should be improved, especially in some parts of the manuscript. It is suggested to seek assistance of a native speaker or a person with comparable competence.

More in Detail

Title:

The term “retinopathy phenotypes” (again used in the text) may be misleading. Readers probably might expect a clear image description how these phenotypes might look like. Instead of this, not a static image but a classified type of retinopathy developmental within the first six months of the study was the basis of defined classification used here. Therefore I would suggest another term. By the way is the orthography of the title not correct. This stand pars pro toto for other deficits in language competence which are only partially mentioned in this review.

Abstract

First Page, First sentence (line 14): ”the” should be deleted

Next line: “different diabetic individuals” – individuals are per definition different. So better delete “different”.

The sentence in line 23 “Twenty-seven eyes……” is not understandable in the present form.

Generally abstracts should be understandable without the need to read the whole manuscript. Mentioning phenotypes A to C in cannot be understood by readers without additional information.

There is no conclusion in the abstract.

I would prefer and suggest a clear cut structure of the abstract text e.g. following the typical headlines of “Purpose” or “Introduction”, “Methods”, “Results” and “Discussion” or “Conclusion”.

Methods

The authors mentioned that their prospective study was registered under NCT03010397 in ClinicalTrials.gov. Checking the published data in ClinicalTrial indicated some discrepancies. The “Actual Study Start Date” (per definition the data of enrolment of the first patient in a prospective study) is mentioned to have been in December 2016. The Actual Primary Completion date was reported as January 11, 2017. This is surprisingly short, but should stand there as it is. The “Actual Study Completion Date” was reported to have been on March 6, 2019. This data is at maximum only 2 years and 3 months after “Acutal Study Stard Date”. How can it be that the study reports on 5 years of follow-up? These five year would not be finished for any of the patient to date!

It is mentioned that a maximum HbA1c of 10 % was set as an inclusion criterion. The authors should state for what period of time before inclusion the maximum was defined or whether this was a referring to a single point measurement.

Since this was a prospective observation the authors should explain the calculation of the power of the study and the evaluation of the necessary number of patient enrolled.

In the ClinicalTrial.gov -registry of this study (NCT03010397), there is 1 primary endpoint and 8 secondary endpoints mentioned. All secondary endpoints are projected to have final data acquisition at 5 years (the time point that is reported here). Since the whole set of data would clearly give a much more thorough description of the analyzed follow-up the result of the whole setup should be described.

The authors report that “any other systemic disease that could affect the eye” was an exclusion criterion. This sounds surprising, since it would mean that any systemic vascular disease or all patients with arterial hypertension would have been excluded. If so, it would be surprising how the authors could recruit such a number of patients. The following sentence, concerning arterial pressure indicates that the exclusion criteria (..any other disease….) were not applied in the mentioned way. There is doubt that the range of accepted systolic and diastolic blood pressure was adequate. The setting would have allowed a patient with a blood pressure of 210/120 to be included. This of course cannot be regarded as an acceptable range.

Page 5 – Line 5: The reading center grading DR severity should be identified.

Page 5 – Line 107: “The endpoints…”: It should be mentioned that these were secondary endpoints of the main study.

The resolution of the used camera for colour fundus photography should be described. The same applies to the RetmarkerDR system.

Details on the CRT measurements should be mentioned. Was it an automatically evaluated focal point measurement? Or was it the average of a central field e.g. 1 mm taken?

Page 4 – Line 81: DPT is unusual. Either Dpt, dpt or D seems more appropriate. All abbreviations should be explained somewhere in the manuscript.

Discussion

Page 15 – lines 280-283: On the background of the presented data and previous literature on diabetic retinopathy it seems not justified to conclude that “CIME and CSME are not similar in their development and evolution and should not be mixed…”. Surely there are differences between CIME and CSME – trivially in their morphological appearance. On the other hand there are so many indicators for similarities of the pathogenesis that such a fundamental division between both would need much more arguments then those presented.

With regard to page 15 (lines 284-294) the question arises whether the types (A, B, or C) remain identical over time. The authors should provide data of how many cases switched to other categories than the initial one. If type A presents neurodegeneration and type B and C not it seems that type A probably will never switch to type B or C (because neurodegeneration cannot disappear). This should be discussed.

Page 16 -Lines 295-300: There is no proof that neurodegeneration develops “independently of microvascular changes”. At least the data presented is not sufficient to underline this argument. Generally, the data presented on neurodegeneration have to be discussed critically. To measure and compare the RNFL thickness very defined topologic positions have to be selected that allow comparison. A gold standard of RNFL measurement is to evaluate data in ring-like peripapillay or papillary position. There are many methodological arguments in favour of such an approach and the authors should have added this standard technique in order to discuss neurodegenerative effects. The statement that “these findings confirm that different disease pathways predominate in different patients” cannot be underlined and seems not be justified by the data presented.

The study is lacking information on the quality of diabetes control and blood pressure control during the 5-years follow-up period. Therefore it cannot be excluded that the quality of care in groups A, B and C was identical or was not of relevance for the follow-up development.

References

The list of references does not cite according to usual standards. 17 of the 18 citations are incomplete. Given the fact that there are seven members in the team of authors this is rather surprising. Obviously none of the authors has carefully checked the final version of the manuscript.

Author Response

Answers to Reviewer #2

Comments and Suggestions for Authors

In their manuscript entitled “Different Retinopathy phenotypes in Diabetes Type 2 show different five-year progression risks for developing macular edema and proliferative retinopathy” Marques et al. report on results of a prospective follow-up study in which the authors observed that the initial development of diabetes, classified into three groups, correlates with further long-term disease progression.

Results are part of the study with 9 mentioned endpoints (ClinicalTrials.gov). Here only few results of the whole study are described. The authors should give information on the other endpoint data.

 We have now included a short phrase, in the methods section, stating that data was also collected for OCT-thickness in outer and inner-rings, OCT-angiography, and OCT-leakage, but due to space limitations and focus chosen for this report, this data will subject of future report (lines 170-172) “Data was also collected for OCT analysis of retinal segmentations in the inner and outer-rings, OCT-leakage (17) and OCT-angiography (18) (performed in the last visit), which will be the subject of future reports”.

Grammar, style and linguistic performance of the manuscript should be improved, especially in some parts of the manuscript. It is suggested to seek assistance of a native speaker or a person with comparable competence.

The grammar and linguist styles were reviewed carefully by the authors.

More in Detail

  • Title:

The term “retinopathy phenotypes” (again used in the text) may be misleading. Readers probably might expect a clear image description how these phenotypes might look like. Instead of this, not a static image but a classified type of retinopathy developmental within the first six months of the study was the basis of defined classification used here. Therefore I would suggest another term. By the way is the orthography of the title not correct. This stand pars pro toto for other deficits in language competence which are only partially mentioned in this review.

The title has been simplified to “Retinopathy phenotypes in type 2 diabetes with different risks for macular edema and proliferative retinopathy”. The basis for the categorization of the different phenotypes has now been introduced in the beginning of the abstract.

  • Abstract
  • First Page, First sentence (line 14): ”the” should be deleted.
  • Next line: “different diabetic individuals” – individuals are per definition different. So better delete “different”.

Both suggestions were taken in account and the words “the” and ”different” were removed as suggested

  • The sentence in line 23 “Twenty-seven eyes……” is not understandable in the present form.

The expression has been clarified and replaced by “Twenty-seven eyes (16%) developed either CSME (14), CIME (10) or PDR (4), with one eye developing both CSME and PDR”.

  • Generally abstracts should be understandable without the need to read the whole manuscript. Mentioning phenotypes A to C in cannot be understood by readers without additional information.

We have rewritten the abstract to make it more understandable and clearer to the readers.

  • There is no conclusion in the abstract.

We have now added a sentence stating the conclusion In conclusion, phenotype C identified eyes at higher risk for development of vision-threatening complications, particularly CSME and PDR. In contrast, phenotype A identifies eyes at very low risk for vision-threatening complications.

  • I would prefer and suggest a clear cut structure of the abstract text e.g. following the typical headlines of “Purpose” or “Introduction”, “Methods”, “Results” and “Discussion” or “Conclusion”.

The typical headlines suggested are not accepted by the journal, as state in the “Instructions for authors”. We made an effort to rewrite the abstract taking in account the journal directives and your suggestions.

  • Methods
  • The authors mentioned that their prospective study was registered under NCT03010397 in ClinicalTrials.gov. Checking the published data in ClinicalTrial indicated some discrepancies. The “Actual Study Start Date” (per definition the data of enrolment of the first patient in a prospective study) is mentioned to have been in December 2016. The Actual Primary Completion date was reported as January 11, 2017. This is surprisingly short, but should stand there as it is. The “Actual Study Completion Date” was reported to have been on March 6, 2019. This data is at maximum only 2 years and 3 months after “Acutal Study Stard Date”. How can it be that the study reports on 5 years of follow-up? These five year would not be finished for any of the patient to date!

The prospective study registered under NCT03010397 in ClinicalTrials.gov is a continuation of 2 previous shorter observational prospective clinical studies from AIBILI (NCT01145599 and NCT01607190), that used the same inclusion criteria and methodologies.

  • It is mentioned that a maximum HbA1c of 10 % was set as an inclusion criterion. The authors should state for what period of time before inclusion the maximum was defined or whether this was a referring to a single point measurement.

Hemoglobin A1c values higher than 10% were an exclusion criteria, using the values collected from the first visit (baseline, v0) and the six-month visit. In summary, HbA1c values collected during the first 6 months of the study should not be higher than 10%. This is now clarified in the manuscript (lines 79-80).

  • Since this was a prospective observation the authors should explain the calculation of the power of the study and the evaluation of the necessary number of patient enrolled.

The calculation of the sample size was based on previous studies (6–9), where the existence of three distinct phenotypes of DR progression in T2D was detected. Their incidence on the DR population was approximately 50% for phenotype A, 30% for phenotype B and 20% for phenotype C. Individuals with phenotypes B and C were shown to be at higher risk of developing CSME treatment, with 11% of phenotype B and 30% of phenotype C developing CSME in a 2-year interval. In order to have at least 5 eyes from phenotype B and 10 eyes from phenotype C developing CSME treatment, a total of 200 eyes were considered to be needed for the study.

Information on this subject has been now included in the text lines 119-126.

  • In the ClinicalTrial.gov -registry of this study (NCT03010397), there is 1 primary endpoint and 8 secondary endpoints mentioned. All secondary endpoints are projected to have final data acquisition at 5 years (the time point that is reported here). Since the whole set of data would clearly give a much more thorough description of the analyzed follow-up the result of the whole setup should be described.

This point as previously been addressed. Due to the space limitations the results collected for OCT-thickness in inner and outer ring, OCT-Leakage and OCT-Angiography will be subject of future reports.

  • The authors report that “any other systemic disease that could affect the eye” was an exclusion criterion. This sounds surprising, since it would mean that any systemic vascular disease or all patients with arterial hypertension would have been excluded. If so, it would be surprising how the authors could recruit such a number of patients. The following sentence, concerning arterial pressure indicates that the exclusion criteria (..any other disease….) were not applied in the mentioned way. There is doubt that the range of accepted systolic and diastolic blood pressure was adequate. The setting would have allowed a patient with a blood pressure of 210/120 to be included. This of course cannot be regarded as an acceptable range.

The sentence has been rewritten to replace the expression used “…any patient comorbidity, likely to affect the eye and not related with diabetes or cardiovascular disease.”

  • Page 5 – Line 5: The reading center grading DR severity should be identified.

The ETDRS grade classification was performed in an experienced reading centre, Coimbra Ophthalmology reading Center (CORC). This information was added to the manuscript.

  • Page 5 – Line 107: “The endpoints…”: It should be mentioned that these were secondary endpoints of the main study.

The word “endpoints” has now been corrected to “the outcomes”

  • The resolution of the used camera for colour fundus photography should be described. The same applies to the RetmarkerDR system.

The resolution of the fundus camera for CFP and Retmarker system was 3596x2448 pixels. This information is now stated in manuscript, in the respective topic in the methods section.

  • Details on the CRT measurements should be mentioned. Was it an automatically evaluated focal point measurement? Or was it the average of a central field e.g. 1 mm taken?

The CRT measurements were obtained directly from the OCT equipment and refer to an average of the central field in 1mm. This information is now stated in manuscript

  • Page 4 – Line 81: DPT is unusual. Either Dpt, dpt or D seems more appropriate. All abbreviations should be explained somewhere in the manuscript.

It is now referred as D.

  • Discussion
  • Page 15 – lines 280-283: On the background of the presented data and previous literature on diabetic retinopathy it seems not justified to conclude that “CIME and CSME are not similar in their development and evolution and should not be mixed…”. Surely there are differences between CIME and CSME – trivially in their morphological appearance. On the other hand there are so many indicators for similarities of the pathogenesis that such a fundamental division between both would need much more arguments then those presented.

The discussion has been rewritten (lines 296-301) “These findings suggest that different factors underly the development of CIME and CSME. The association of increased values of MAT with CSME points to the potential relevance of capillary closure and ischemia in the process of CSME development. Therefore, CIME may not be necessarily a predictive factor for CSME (20). It is also an argument against relying entirely on OCT metrics to identify DME (4).”

  • With regard to page 15 (lines 284-294) the question arises whether the types (A, B, or C) remain identical over time. The authors should provide data of how many cases switched to other categories than the initial one. If type A presents neurodegeneration and type B and C not it seems that type A probably will never switch to type B or C (because neurodegeneration cannot disappear). This should be discussed.

In response to the question of the phenotypes A, B and C remaining identical overtime, a sentence has now been included in lines 322-325. ” Noteworthy, the eyes with phenotypes A and B remained mostly stable through the 5-year follow-up, with only 2 eyes with phenotype A (3%) and none with phenotype B presenting 2-step ETDRS grade worsening, whereas phenotype C showed 2-step ETDRS worsening in 13 eyes (23%).

 Phenotype characterization was not performed at the end of the study and therefore it is not possible to know whether the initial phenotype characterizations remained the same in the course of the disease.

  • Page 16 -Lines 295-300: There is no proof that neurodegeneration develops “independently of microvascular changes”. At least the data presented is not sufficient to underline this argument. Generally, the data presented on neurodegeneration have to be discussed critically. To measure and compare the RNFL thickness very defined topologic positions have to be selected that allow comparison. A gold standard of RNFL measurement is to evaluate data in ring-like peripapillay or papillary position. There are many methodological arguments in favour of such an approach and the authors should have added this standard technique in order to discuss neurodegenerative effects. The statement that “these findings confirm that different disease pathways predominate in different patients” cannot be underlined and seems not be justified by the data presented.

We agree with the reviewer comments and deleted the controversial explanations given for “neurodegeneration findings”. The discussion has been changed taking in consideration the comments made by reviewer.

  • The study is lacking information on the quality of diabetes control and blood pressure control during the 5-years follow-up period. Therefore it cannot be excluded that the quality of care in groups A, B and C was identical or was not of relevance for the follow-up development.

The levels of HbA1c and blood pressure were collected and registered in all study visits, although not analysed in the context of the work presented in this manuscript. We just focused on the HbA1c levels in the first 6 months, that should be lower than 10%, for inclusion criteria.

Our study was performed in collaboration with the Central University Hospital of Coimbra, where the patients are followed. Hence, the healthcare provided was the same for all individuals included in the study.

  • References

The list of references does not cite according to usual standards. 17 of the 18 citations are incomplete. Given the fact that there are seven members in the team of authors this is rather surprising. Obviously none of the authors has carefully checked the final version of the manuscript.

We have used the Mendeley Software to organize the reference list, using the style of the JCM. For proper revision, we have revised all the references presented in this version of the manuscript.

Reviewer 3 Report

Marques et al. present an interesting report that extends their group’s previous observations of identifying different risk of eye complications in phenotypes of diabetic retinopathy classified by different microaneurysm turnover or subclinical macular edema. Their effort of performing long-term prospective studies is highly valuable and appreciated.

Specific comments:

p.6 line 149: The segmentation procedure of OCT images should be described in more detail. There is no specific information about this in the given reference. Was segmentation really performed in OCTA structural data? The scan pattern in OCTA would be different than described.

p.11, Table 2: It would be interesting to see the patient characteristics at the end of the study, i.e. if there were changes in systemic and ocular parameters. Did all patients remain in their basic phenotype classification? In addition, some information about medical treatment may be of interest to the reader.

p.11, line 220-224: The authors claim that phenotype A shows neurodegeneration in the central subfield of OCT. This statement seems not really supported by the data. When neurodegeneration occurs, this should affect the area with more neuronal tissue, i.e. the ring segments of the ETDRS grid which contain more cell bodies and fibers in the inner layers. Higher values in the central field of phenotype B more likely represent swelling due to subclinical edema. GCL and RNFL swelling in all ETDRS subfields has been shown in patients with DME. In addition, the comparison with the healthy control group seems not valid since this group was considerably younger (mean age 47.6) and therefore not matchable. Also, data of individual OCT subfields have not been presented in the given reference and are not accessible. A device-specific normative reference database of retinal layer thickness values still needs to be established. I would strongly recommend to omit the statements about neurodegenerative findings in this subcohort.

p.12, line 245: If I understand the statistical model correctly, rather an increase in GCL+IPL thickness increased the odds of developing macular edema.

p.13, Figure 2: This figure does not add much information to the text and could be removed.

p.15, line 292-300: see above comment for p.11

Minor remarks:

p.1, line 24: Please rephrase for clarity. It reads as twenty-seven eyes developed CSME.

p.2, line 52: Please add the references for the 2-year follow-up studies.

p.4, line 69: the sentence needs change to be understood correctly. I suggest ‘…development of PDR or diabetic macular edema (DME), considering both clinically significant macular edema (CSME) and center involved macular edema (CIME).’

p.5, line 114: How were new vessels detected? If this was done only using 7-field fundus photography peripheral new vessels could have been missed.

Author Response

Answers to Reviewer #3

Comments and Suggestions for Authors

  • Marques et al. present an interesting report that extends their group’s previous observations of identifying different risk of eye complications in phenotypes of diabetic retinopathy classified by different microaneurysm turnover or subclinical macular edema. Their effort of performing long-term prospective studies is highly valuable and appreciated.

  • Specific comments:
  • 6 line 149: The segmentation procedure of OCT images should be described in more detail. There is no specific information about this in the given reference. Was segmentation really performed in OCTA structural data? The scan pattern in OCTA would be different than described.

The detailed information was now included in the methods section (lines 152-162) “Retinal layers segmentation for layer thickness calculation was performed on the structural OCT or OCTA (in the last visit) using the segmentation software implemented by AIBILI (16). In the first step of our implementation, the algorithm segments large voxel intensity variations, i.e, vitreous to Retinal Nerve Fiber Layer (RNFL), Inner Segment (IS) to Outer Segment (OS) and Retinal Pigment Epithelium (RPE) to choroid interfaces. The two outer most interfaces are then used as boundaries to find the remaining inner retinal layer interfaces. All surface interfaces are then smoothed by cubic splines. This method is able to segment the OCT volume into seven retinal layers, namely, RNFL, Ganglion Cell Layer and Inner Plexiform Layer (GCL+IPL), Inner Nuclear Layer (INL), Outer Plexiform Layer (OPL), Outer Nuclear Layer and Inner Segment (ONL+IS), OS and RPE.”

  • 11, Table 2: It would be interesting to see the patient characteristics at the end of the study, i.e. if there were changes in systemic and ocular parameters. Did all patients remain in their basic phenotype classification? In addition, some information about medical treatment may be of interest to the reader.

The classification of phenotype was only performed at 6 months visit, after MAT and CRT assessment. No phenotype classification was performed at the end of the study.

  • 11, line 220-224: The authors claim that phenotype A shows neurodegeneration in the central subfield of OCT. This statement seems not really supported by the data. When neurodegeneration occurs, this should affect the area with more neuronal tissue, i.e. the ring segments of the ETDRS grid which contain more cell bodies and fibers in the inner layers. Higher values in the central field of phenotype B more likely represent swelling due to subclinical edema. GCL and RNFL swelling in all ETDRS subfields has been shown in patients with DME. In addition, the comparison with the healthy control group seems not valid since this group was considerably younger (mean age 47.6) and therefore not matchable. Also, data of individual OCT subfields have not been presented in the given reference and are not accessible. A device-specific normative reference database of retinal layer thickness values still needs to be established. I would strongly recommend to omit the statements about neurodegenerative findings in this subcohort.

Indeed, this is a point that we want to further explore in the data collected in the study population. We thank the reviewer for the suggestion. We have removed these sentences.

  • 12, line 245: If I understand the statistical model correctly, rather an increase in GCL+IPL thickness increased the odds of developing macular edema.

Yes, according to our model GCL+IPL thickness is associated with increased risk for both CSME and CIME. We have used a multivariate model of block entry that is based all variables that had, in the univariate analysis, p≤0.1. This method was used to decrease the model complexity and enhance its external variability.

  • 13, Figure 2: This figure does not add much information to the text and could be removed.

We believe that the distribution of outcomes on the 5-year period is relevant. It indicated thar longer duration of disease is not a factor for the development of vision-threatening complications

  • 15, line 292-300: see above comment for p.11

We have taken the reviewer comment in consideration, and the statements discussing neurodegeneration were corrected and/or removed. Please see the changes made in the text (lines 309-311). We have just kept these statements in our discussion:” Phenotype A is also the phenotype associated with increased thinning of the GCL+IPL in the central subfield. This observation contrasts with the absence of changes in the GCL+IPL in the inner ring, outside the central subfield.”

  • Minor remarks:
  • 1, line 24: Please rephrase for clarity. It reads as twenty-seven eyes developed CSME.

It has been rephrased

  • 2, line 52: Please add the references for the 2-year follow-up studies.

The references were included:

  1. Nunes S, Ribeiro L, Lobo C, Cunha-Vaz J. Three different phenotypes of mild nonproliferative diabetic retinopathy with different risks for development of clinically significant macular edema. Investig Ophthalmol Vis Sci. 2013;10(54(7)):4595–604.
  2. Ribeiro L, Bandello F, Tejerina AN, Vujosevic S, Varano M, Egan C, et al. Characterization of retinal disease progression in a 1-year longitudinal study of eyes with mild nonproliferative retinopathy in diabetes type 2. Investig Ophthalmol Vis Sci. 2015;
  3. Ribeiro ML, Nunes SG, Cunha-Vaz JG. Microaneurysm turnover at the macula predicts risk of development of clinically significant macular edema in persons withmild nonproliferative diabetic retinopathy. Diabetes Care. 2013;36(5):1254.
  4. Ribeiro L, Pappuru R, Lobo C, Alves D, Cunha-Vaz J. Different Phenotypes of Mild Nonproliferative Diabetic Retinopathy with Different Risks for Development of Macular Edema (C-TRACER Study). Ophthalmic Res. 2018;59(2):59–67.

  • 4, line 69: the sentence needs change to be understood correctly. I suggest ‘…development of PDR or diabetic macular edema (DME), considering both clinically significant macular edema (CSME) and center involved macular edema (CIME).’

Suggestion accepted. This change was made

  • 5, line 114: How were new vessels detected? If this was done only using 7-field fundus photography peripheral new vessels could have been missed.

New vessels were detected by clinical examination

Round 2

Reviewer 2 Report

The response to the first comment just states that further parameters were examined in this study. The response, that due to “space limitations and focus chosen for this report, this data will be subject of future report” is not convincing and satisfying. The data no presented is essential for the interpretation of results and therefore should have been presented together with the set chosen here.

Although “The basis for the categorization of the different phenotypes has now been introduced in the beginning of the abstract” the reader of the abstract still does not know what types A to C stand for.

Concerning the aspects of study registration: It should have been made clear that this is an extension of two previously finished studies and that the patients of these previous studies were extended into this protocol. Further the NTC numbers of the previous study should have been mentioned. More in detail, it is obviously when reading the three descriptions, methods and selection criteria in ClinicalTrial it is obvious that the three studies were not identical as mentioned, albeit considerable overlapping. It is good research practise in cases of extended studies to mention explicitly that certain patients had been part of a previous study and that results of these cohorts had been published under definitely described and listed publications. And the reader askes how the 212 individuals were selected from the larger cohorts of the previous studies.

The question concerning the resolution of the camera did not address the number of pixels of the chip in the camera. More interesting is the question how the resolution of retinal structures or an equivalent of a retinal image might. This is important to know with regard to the resolution and cut off in the detection of small microaneurysms or more precisely of red spots on the fundus.

Discussion of page 15 – lines 280-283 (first version): Presentation of OCTA, as suggested, might perhaps be a wonderful support to the argumentation.

Missing data on the quality of diabetes care seems to be an essential control parameter and I would suggest to collect these data retrospectively from the caring centres.

Author Response

In response to the comments and suggestions received from reviewer number 2 we have made the following corrections, as requested:

Reviewer 2

The response to the first comment just states that further parameters were examined in this study. The response, that due to “space limitations and focus chosen for this report, this data will be subject of future report” is not convincing and satisfying. The data no presented is essential for the interpretation of results and therefore should have been presented together with the set chosen here.

We agree that the not data presented would be of value. Exams like OCTA were only performed in the last visits of the study and not in all patients, and therefore it is not possible to correlate the with outcome development, which is the aim of this report.

Although “The basis for the categorization of the different phenotypes has now been introduced in the beginning of the abstract” the reader of the abstract still does not know what types A to C stand for.

We have now proceed to characterized the 3 phenotypes in the beginning of the abstract, as requested (line 11-14) “Our group reported that three diabetic retinopathy (DR) phenotypes characterized:  A by low microaneurysm turnover (MAT<6) and normal central retinal thickness (CRT); B low MAT(<6) and increased CRT, and C high MAT(≥6), present different risks for development of macular edema (DME) and proliferative retinopathy (PDR).”

Concerning the aspects of study registration: It should have been made clear that this is an extension of two previously finished studies and that the patients of these previous studies were extended into this protocol. Further the NTC numbers of the previous study should have been mentioned. More in detail, it is obviously when reading the three descriptions, methods and selection criteria in ClinicalTrial it is obvious that the three studies were not identical as mentioned, albeit considerable overlapping. It is good research practise in cases of extended studies to mention explicitly that certain patients had been part of a previous study and that results of these cohorts had been published under definitely described and listed publications. And the reader askes how the 212 individuals were selected from the larger cohorts of the previous studies.

Regarding aspects of study registration, we have now included in the text the NTC numbers of the previous studies and respective publications (lines 64-67): “This study is an extension of three previous studies (NCT0114599(9), NCT01607190(7) and EudraCT2012-001200-38(11)). The patients in the study here reported were included according to specified inclusion and exclusion criteria and they were followed for a period of 5 years…”

The question concerning the resolution of the camera did not address the number of pixels of the chip in the camera. More interesting is the question how the resolution of retinal structures or an equivalent of a retinal image might. This is important to know with regard to the resolution and cut off in the detection of small microaneurysms or more precisely of red spots on the fundus.

The Retmarker makes a relative measurement of MAT by comparing two photographs and reports on changes overtime, therefore algorithm pics only differences between the red-spots within the two photographs taken in a set-time interval in the same patients using the same equipment.

Discussion of page 15 – lines 280-283 (first version): Presentation of OCTA, as suggested, might perhaps be a wonderful support to the argumentation.

As already mentioned, OCT-A was performed in the last the last two visits of the study and not in all patients, which do not allow a correlation with the outcomes. A sentence has now been included in the manuscript to explained the non-inclusion of OCT-A data in this report (line 176-178): “A OCT-A was performed only in the last two annual visits and the data collected could not be analysed in correlation with the outcomes CSME, CIME or PDR.”

Missing data on the quality of diabetes care seems to be an essential control parameter and I would suggest to collect these data retrospectively from the caring centres.

Although we agree that a follow-up of the metabolic parameters would be of interest, the aim of this study was how baseline values may predict vision-threatening complications in people T2D and not on the effect of metabolic control during the study. 
